# The Head Coach as a Coach Developer: A Coach Education Case Study inside a World Champion Futsal Team

Ana Gracinda Ramos [1,*] , Carla Valério [2] and Isabel Mesquita [1]

1   Centre for Research, Education, Innovation, and Intervention in Sport (CIFI2D), Faculty of Sport,
    The University of Porto, 4200-450 Porto, Portugal; imesquita@fade.up.pt
2   Faculty of Education, Southern Cross University, Gold Coast, QLD 4225, Australia; carla.valerio@scu.edu.au
*   Correspondence: agramos@fade.up.pt

**Abstract:** As teacher education, coach education is also a continuous and unfinished path by nature. This study explored how the head-coach (HC) of a world champion futsal team worked as a coach developer of their assistant-coaches (AC). Specifically, the main pedagogical strategies used, how they were applied, and impacted on the perceived professional development of their staff members were investigated. An interpretative case-study was adopted since it enables an in-depth investigation of the complexity and uniqueness of this particular technical staff in educational context. Participants included four experienced coaches, one HC and three ACs. Data was gathered through semi-structured interview method and analyzed by thematic analysis. Findings highlighted an intentional structuration of sequence and timings of the pedagogical tasks and activities assigned by the HC to ACs. Evidence emphasized the (i) vision of mistakes as learning opportunities, (ii) fostering commitment of ACs to enhance team performance, and (iii) space to plan and lead in practical contexts as the main strategies adopted by the HC, who also considered reflective skills as a paramount competency and pedagogical strategy in coach education. From ACs' perspective, these strategies largely impacted their personal and professional growth. Findings could guide the re-design of future coach education programs.

**Keywords:** coach education; pedagogy; teaching-learning; professional development; situated and experiential learning

## 1. Introduction

As in teacher education, coach education is also a continuous and unfinished path by nature [1]. To date, many pertinent investigations have studied how coach education programs could be implemented and adapted, with a particular focus of analysis being placed on the initial education of novice coaches (see [2,3]). However, despite the career stage, professional and social development of coaches tends to be a continuous, progressive and situated process [4]. Indeed, to achieve expertise throughout their careers, coaches must learn, transform their lived experiences into knowledge, and apply such knowledge in practice (i.e., coaches' professional development) [5,6].

While the education of novice coaches frequently occurs through formal and non-formal environments (i.e., organized and systematic educational activity that occurs inside and outside the framework of the formal system, respectively [7]), more experienced coaches tend to learn from sharing experiences with their peers in informal environments (e.g., unplanned conversations with peers or learning through share on-the-field experience) [8,9]. For this reason and given the constant interaction within the technical staff (head coach, assistant coaches, scouters, etc.) in real contexts of practice, the coaches' professional development could—and perhaps should—be highlighted inside the technical staffs. Thus, a step forward in coach education research is investigating how a technical staff works in high sport performance contexts. By understanding how the head coach

works to raise the sport performance of his or her staff, such investigations could provide useful insights on what the good practices that sustain a coach's professional development over time are (and not exclusively in the early stages of a coaching career), informing the design of coach education programs.

Technical staffs are privileged learning contexts, given their practical and situated nature which enables a meaningful interpretation of all decisions made [10,11]. Notwithstanding, so that the long-term coach education process becomes possible and real, the head coach, as the leader of the technical staff, needs to acknowledge the importance of the continuous education of their peers (i.e., assistant coaches), embracing the role of coach developer [12]. The designation of coach developer (CD) has been conceptualized as an umbrella term that includes several roles such as educator, leader, facilitator, or evaluator [12]. Specifically, CDs are recognized as informed experts in coaching practice, skilled facilitators of coaches' learning, and educators responsible for mentoring and challenging their peers through a wide range of structured activities [13,14]. As portrayed in the recent systematic review of Jones et al. [15], the studies focusing on CDs have investigated who CDs are, what they do, and how they do it. However, such investigations have been conducted inside formal coach education programs designed to train novice coaches, such as those in [16–18]. Currently, a theoretical step forward should be taken through in-depth investigations on how the head coaches, within an authentic practical context, could work as CDs inside their staff and how such pedagogical intervention could impact the professional development of their peers. Furthermore, from a practical standpoint, by empowering head coaches to work as CDs, many issues currently reported on coach education programs (e.g., financial state, facilities available, or the excessive uncontextualized theory of the learning contents [9,19]) could be resolved.

The teaching-learning process inside coach education must include three pillars: doing (practical experience), thinking (reflection experience), and interpreting (contextual and social experience) [4,20]. Grounded in experiential and situated learning, these pillars support the transformation of coaches' experience into knowledge and skills through a process of reflection that is embedded in a particular context and culture, thereby becoming inseparable from social practice [10,21,22]. Within the scope of experiential and situated learning, studies have mainly investigated the feasibility of coach education programs, namely the effectiveness of their learning contents and the impact of their vocational training (e.g., [23–25]).

By working together in the same environment, the CD (i.e., head coach) gains the advantage of effectively understanding the historical and situated context of practice, offering the opportunity to better appropriate their pedagogical intervention (i.e., pedagogies and strategies used, and activities suggested) according to the characteristics of the persons involved. Despite the potential contribution of head coaches as CDs of their assistant-coaches, thus far, there remains a gap in the literature on how the CD might potentiate the "learning by doing" of their staff through diverse strategies and activities directly related to their daily responsibilities (e.g., planning of training sessions and scouting). Notably, in countries where coach education has not been systematically organized, the investigation of how coaches in advanced stages of their careers contribute to the ongoing education of their peers inside the same sportive structure (i.e., clubs or federations) acquires an additional significance [9,19]. Given the novelty of this theme, as well as the individual and contextual features of any learning process, an in-depth examination of the teaching-learning context is required. In this regard, interpretative case studies provide valuable insights and well-informed understandings of coaching experiences, supporting the development of new insights that may guide the ongoing professional development of coaches over time.

Within a high-level sport performance context, and through an interpretative case study, this study aims to explore how the head coach of a world champion futsal team worked as a coach developer of their technical staff (i.e., assistant coaches). Specifically, inside this singular learning and performance context, we intended to investigate the

main pedagogical strategies used, how they were applied, and their impacts on their staff members perceived professional development.

## 2. Materials and Methods

### 2.1. Study Design

An interpretative case study [26] was used to examine the pedagogical strategies implemented by the head coach and his or her impact on promoting the professional development of their technical staff. Specifically, a case study was adopted since it enabled an in-depth investigation of the complexity and uniqueness of this particular technical staff in a real-life context, as well as understand the daily activities, routines, practices, and meanings for all those involved [27]. An interpretative paradigm and a qualitative approach [28,29] were used to investigate how the head coach worked as a coach developer intending to stimulate the coach education and professional development of their assistant coaches. Thus, through a case study design, we sought to interpret and explore how and why the head coach tailored their pedagogical intervention (i.e., adapted the strategies used and the activities suggested). In this line, the study adhered to the tenets of ontological relativism and epistemological constructionism [30], which outlines the occurrence of multiple realities toward a personal truth, thereby assuming that knowledge is conceived as a construction dependent on the context and personal experiences. This approach holds the utmost potential to capture, from a practical context of examination, how different formative strategies are applied in ongoing fashion and their impact on the professional development of coaches over time.

### 2.2. Context and Participants

Aligned with the premises of a single case study design, the participants were selected because they encompassed a set of particularities [26], namely (1) the relationship among the head coach and his first assistant coach (AC1), since both worked together over the last 10 years, and the head-coach was, many years ago, a player of his current assistant coach. Years later, both also worked together as assistant coaches on the same technical staff. (2) Since the beginning of working with the national team, the change of roles between head and assistant coach was constant (i.e., the head coach of the under-21 national team performs the role of assistant coach of the under-17 national team, and vice versa). (3) Many members (i.e., assistant coaches) were added over the last 10 years to the technical staff of the national team without any withdraws over the last 11 years. (5) All the members that integrated the technical staff over the years performed the roles of head and assistant coaches for youth national teams (e.g., AC1 is the head coach of the under-17 national team and an assistant coach of the under-19 national team). Finally, (5) regardless of the head coach roles or federative responsibilities, all the members work as assistant coaches of the male and female senior national teams with field responsibilities (scouting, leading practice tasks during training sessions, etc.). Over the last 10 years, this technical staff won two European championships, one university world championship, and one world championship at the senior level. Moreover, the under-19 national team won one European championship.

The head coach (HC) who integrated this study holds a degree in physical education and a master's degree in high-performance sports training. After completing a career as a football goalkeeper, his journey as a coach began. The HC took on the role of national coach in 2010, and since then, he was elected the Best National Futsal Coach in the World four times. Three assistant coaches participated in this study (i.e., AC1, AC2, and AC3). AC1 holds a master's degree in physical education, majoring in primary and secondary education. AC1 started working with the HC in 1996, performing the role of assistant coach of the university's national teams. His connection to the football federation also began in 2010. AC2 holds a degree in physical education and sports with a complementary option of football and debuted as a futsal coach in the 2001–2002 season as an assistant coach. The 2012–2013 season marked the beginning of his involvement in the technical staff of

the national futsal team. AC3 holds a degree in physical education, a master's degree in physical education, and Level III of the futsal coaching course. AC3's connection to the football federation began in 2018.

*2.3. Data Collection*

Data were gathered through a semi-structured interview method [31,32], with each coach being interviewed on one occasion and separately (i.e., different weeks). A semi-structured interview script was developed and refined through a pilot interview conducted with assistant coaches (n = 2) with similar sports backgrounds but less international experience. Such pilot interviews enabled minor but pertinent refinements regarding question clarity. The guide of the semi-structured interview was flexible, enabling an open-ended approach that explored coaches' thoughts, feelings, and convictions about their professional development, namely its importance and how it occurred inside the technical staff (i.e., pedagogical strategies used by the CD) [33]. To understand the pathway of each coach and how their professional relationship was developed, the first stage of the interview included questions about coaches' careers (e.g., for how long? In what sporting contexts? Training, senior, or club: which have you coached so far?). Then, a set of questions was targeted to explore assistant coaches' current personal and professional competencies (e.g., during exercise or training sessions, do the ACs assume full leadership in instruction, almost like inverted roles, and why?). Finally, it was explored how the HC and assistant coaches mutually influence their professional development (e.g., on the other hand, do you feel that you have contributed to the professional growth of your AC? How and in what aspects? Can you give me an example?).

The first participant interviewed was the HD to gather an overall perception of the working routines established, the roles performed by each of the staff members, and the pedagogical strategies used to enrich the coach education of their peers. The three assistant coaches were interviewed three weeks later once we slightly adapted the interview guidelines of the assistant coaches according to the previous information transcribed and examined from the head coach interview. The guide for assistant coach interviews included the same themes and subthemes as the interview guide of the head coach. However, the questions were adjusted according to the specificity of the role performed (e.g., HC or AC). The interviews lasted between 50 and 90 min (M = 70 min). The interviews were conducted in person in a silent room or via video conferencing software (Zoom Video Communications, San Jose, CA, USA, version 5.15). The first and third authors used interview skills (prompts and probing) to encourage participants' descriptions of their living experiences. Thereby, the interviews followed the recommendations of Kvale [34] (i.e., "inter-view"), where both worked together to explore the world of the experience of each participant.

*2.4. Data Analysis*

The first author audio recorded and transcribed verbatim the four interviews. Afterward, thematic analysis [35] was used to examine the data gathered for identifying, examining, and reporting themes within extensive data sets. The six phases characteristic of thematic analysis were completed [36]. As Terry et al. [32] recommended, immersion and familiarization with the data content were first carried out by continuously rereading the interview transcripts. The second phase involved inductive line-by-line open coding to identify the main strategies used, the critical perspectives under investigation for those pedagogical strategies, and how they impact assistant coaches' professional development. Next, the codes were clustered into subthemes and themes. In the fourth stage, the themes were revised to ensure they fit the data content. The fifth stage included working back and forth among data and theory to select and appropriately support the themes identified. Finally, the sixth stage encompassed reporting and writing the results. Throughout this process, an intentional effort was made not to force data to fit preestablished theories

(i.e., inductive approach), seeking renewed insights that could agree with or refute current theoretical tenets [37].

*2.5. Trustworthiness*

The Declaration of Helsinki guidelines were followed, and the study was approved by the first author's institutional research ethics committee. The participants were informed about the scope of this investigation and the possibility to withdraw at any time. Then, informed consent forms were signed. Anonymity was guaranteed through use of pseudonyms.

Qualitative data analysis involves issues of subjectivity in data interpretations that must be acknowledged [38]. To ensure data trustworthiness, several procedures were adopted: (1) the interviews were conducted under an environment of care and impartiality, with the coaches feeling free to express their genuine perspectives [32]. (2) After the first transcription, the first author revised their own transcription by hearing the interviews again. Next, the audio recordings and respective transcription were sent to the second author, who double-checked the writing content [39]. (3) The coaches were systematically questioned about the true meaning of their verbal interventions. Accordingly, following the transcription of semi-structured interviews, the coaches were invited to add, adjust, or delete information, with the aim of clarifying the views that they intended to share (i.e., member checking) [40]. (4) Finally, the second and third authors acted as critical friends and questioned the interpretations made by the first author at each stage [41]. Together, these procedures contributed to raising the study's trustworthiness, concomitantly reducing the individual research bias.

## 3. Results

*3.1. An Intentionally Structured Baseline of Work to Promote Professional Development in Real Educational Contexts*

At the basis of engaging in the process of being a coach developer is the genuine commitment of the HD in the coach education and professional development of his or her staff. For the HC, coaches' professional development is viewed as an ongoing and therefore unfinished learning process that must be nurtured. Furthermore, as highlighted in the following excerpt, a structuration of the work (i.e., tasks and activities) that guides and monitors the coach education process inside the staff is required:

> "*Interviewer (I): How do you do this education or training* [professional development] *with the assistant-coaches?*
>
> *HC: Constantly. The last one was two hours of training, led by me.*
>
> *I: This training is internal, these meetings are internal, led by you...*
>
> *HC: Yes, but we have a series of activities and organization that leads to this.*
>
> *[...]*
>
> *I: But was this structure of activities conceived by you? Or is it something already done before and that you are now implementing?*
>
> *HC: No, it was built by me.*
>
> *I: And what was at the base?*
>
> *HC: Some things were happening, we added them* [activities] *over the years..."*

*#1 HC*

Beyond the set of activities and formal moments, the coach developer also highlighted the importance of all the staff members spending much time together. By working as a coach developer, the HC intentionally builds these moments by scheduling the training camps of youth national teams on the same dates and in the same places. Throughout those moments, there is space for non-formal environments of education. Also, as highlighted

in the last part of the excerpt, the activities are mandatory, emphasizing the demand and accountability requested by all staff members:

> *"HC: I will have all the coaches together 10 times until June because I will always have a female and male training camp in the same place and on the same days. In these training camps we are together, we eat together, we live together, everyone. And on Monday, we have a formal moment, 'Let's look at this part now'* [concerning futsal content]*. But more than this more formal aspect, it is what activities and dynamics we can do as a technical team that are mandatory."*

> *#2 HC*

Despite the obligatory structured activities, they are considered by the assistant coaches to be an important part of the responsibilities inherent in their job. Indeed, these activities are recognized as a method to structure their professional development and coach education instead of tasks that must be merely performed. Specifically, it is acknowledged by the staff that these activities stimulate a deeper understanding of the game, raise the awareness of what "being a coach" means, and enrich the repertoire of problem-solving skills:

> *"I: Do you feel that this form of leadership (e.g., moving from the backstage to the frontline, assigning diverse responsibilities) helped you in your coaching training?*

> *AC1: Oh, definitely! His way of leading, of compelling us. It is not a matter of forcing us, it is part of our tasks and functions, but we are constantly watching games, analyzing certain situations, modifying our documents to improve them, and adapting our game model. All these little things have helped us. They have helped me improve my skills and my knowledge of the game. My knowledge of the game is much greater, I have no doubt about that, than, for example, when I was coaching a team in the 1st league. And this way of leading, giving us tools, inviting us for certain tasks like studying opponents and different opponents, and proposing us for looking to ourselves [. . .] These things lead us to enrich our repertoire and be better coaches. I don't have the slightest doubt about that."*

> *#1 AC1*

Indeed, as expressed next by AC2, such structuration of different tasks and activities enabled the coach education process's organization and played a critical role in raising personal and professional growth. Implicitly, the assessment acquires a formative sense since its utility is in understanding what went right or wrong as well as the reasons behind it. Additionally, the assessment induces a natural reflection that integrates the personal examination of lived experiences and its impact on their peers, stimulating the search for new solutions and renewed insights:

> *"AC2: These are tasks for personal development and organizing the process. Even those who arrive know what they will find and how the process is conducted. The previous reports are done. In the training camp, we do a report with everything that was carried out: its assessment, the assessment of players, and the assessment of staff. The staff is also assessed! And this is all documented.*

> *I: Does it force you to think more and more about the game and get to know it better?*

> *AC2: Yes, without any doubt. It forces us to evolve. It forces us to look for different and new things."*

> *#1 AC2*

What is more, the way the HC intentionally connects and sequences all the activities to promote the structured development of reflection skills, supporting and enriching the professional development of their peers, is highlighted in the following excerpt:

*"AC3: It is all connected. For example, our observations at the weekend are linked with what we want to see and with what our day-to-day reflection was about.*

*I: So, do you feel that reflection is decisive in the coach training process?*

*AC3: Yeah, yeah! Structured reflection. Properly structured."*

*#1 AC3*

*3.2. Pedagogical Strategies for Inducing Professional Development Inside the Staff*

As a coach developer, the HC uses a set of pedagogical strategies such as (1) seeing mistakes as learning opportunities, (2) fostering commitment from every member to raising the team's performance, and (3) providing space to plan and lead field tasks. The reflection skills are considered paramount for the HC and emerge as a pedagogical-didactical strategy (i.e., process-oriented to personal and professional growth) and a competency required for being a coach.

3.2.1. Empowering Learning by Disempowering Mistakes

More than the head coach acknowledging the importance of building learning cultures inside the real practice context, the work as a coach developer entails an important pedagogical strategy: viewing mistakes as learning opportunities. Moreover, as exemplified next, mistakes are considered ways to add a different and complementary vision:

*"I: So, you advocate developing learning cultures in the practice context.*

*HC: Yes, of course.*

*I: Do you think this is what improves coach education?*

*HC: I think it helps. However, [. . .] I often said: 'this is a mistake'.*

*I: And do you allow it?*

*HC: Yes, I do. I do not interfere.*

*I: Why?*

*HC: Because it is the only way for me to let them add what they have, what they see differently. Otherwise, I will be constantly inhibiting it."*

*#3 HC*

In the same manner, inside the collaborative working culture developed, all staff members recognize mistakes as the ingredient that allow moving forward in professional development. As reported by the assistant coaches, even in challenging and exposing situations, this positioning of the head coach (i.e., "you can make mistakes") provides safety and comfort to failing. Hence, assistant coaches are free to be themselves and grow professionally in real-life practice contexts. Instead of weaknesses, mistakes are viewed by all staff members as learning opportunities, especially since their detection is often recognized through reflection:

*"AC1: I am leading an exercise. If it is something very specific that he* [HC] *wants to highlight, he does it. But if I fail and get all mixed up in leading the exercise, he does not stop me.*

*I: In other words, there is always protection.*

*AC1: Always! And that's just the way he is. This is also to avoid this. . .*

*I: To avoid discrediting you in front of the players.*

*AC1: Exactly."*

*#2 AC1*

*"AC3: For example, I believe that if we make a mistake, he will not expose us to it or correct us. Anything. Absolutely sure.*

*I: So?! What?!*

*AC3: Now when we reflect on what happened, we recognized the mistakes that we made"*

#2 AC3

### 3.2.2. Engaging Everyone in the Team's Performance to Promote Professional Development

From the head coach's perspective, the commitment of all the staff members in the tasks serves to increase the team's performance in competition and also as pedagogical strategies for increasing the understanding of the game. Thus, by intentionally inviting their assistant coaches to examine and scout their opponents and presenting all the information gathered to promote collaborative and critical reflection (i.e., staff meetings in which reflexive talks are paramount), the head coach emphasized and supported the notion that everyone is accountable:

*"HC: [...] James and Matthews will analyze the Portugal-Serbia match in the World Cup. No one can send them the analysis that we have already done. They will examine the game and cut the videos. I want to see what their eyes see.*

*I: But does everyone have space to discuss everything with each other? Or not?*

*HC: Everything, everything.*

*I: And why do you encourage this?*

*HC: Because I think the essential is to understand the game. It is understanding what it means to be a coach."*

#4 HC

The assistant coaches' perspective not only underlines the example stated by the head coach but also emphasizes the importance of this pedagogical strategy in making everyone accountable for the team's performance. Also, this feeling is extended to a critique about the common role of assistant coaches inside the technical staff:

*"AC1: Scouting is perhaps the best example. [...] HC normally delivers several matches for we analyze the opponents. For example, two or three games for each one of us. We all video cut these games and collect information in different game moments. Then, we have an initial meeting where everyone presents what we saw.*

*I: And the fact that everyone has that responsibility and exposure. That means moving you to the frontline of the technical team...*

*AC1: This is fundamental for us to feel that we are an important part of the process."*

#3 AC1

Together, these excerpts also emphasized the importance of social learning, namely the building of opportunities for every member to share the same or different viewpoints about the game.

### 3.2.3. Providing Space for Planning and Leading: Learning by Doing

Another pedagogical strategy the head coach highlights is assigning responsibilities for designing and leading drills in practice sessions. Such delegation is used to engage everyone in the process. Also, by providing that kind of opportunity to the assistant coaches, the CD acknowledges what has been learnt so far and the learning contents that need to be addressed further:

*"I: For you, what were the criteria for deciding which and when responsibilities are assigned to your assistant coaches?*

*HC: For me, it is unthinkable that an assistant-coach spend 3-days in the training camp without having a responsibility in which they lead. They must jump to the frontline of*

*the stage somehow. They must feel part of the process. This is very important to me. The content?! No! AC3 must know how to attack, defend, the transition. They all have to know the tactical schemes.*

*I: You empower them so that they can feel committed.*

*HC: A lot! A lot! More than the content by itself. Hiddenly, it is an opportunity for me to understand what they already mastered and the remaining flaws."*

*#5 HC*

The assignment of responsibilities is detailed in the following excerpt by AC1, in which it is noted that the training sessions' planning is built collaboratively by the technical staff but led and managed by the head coach. The assistant coach views this pedagogical strategy as an incredible opportunity not only for them but also to potentiate the performance of the team and players, since it allows the HC focus more on how the practice is unfolding and not exclusively on the dynamics of that specific task (control the time, rotation of players, feedback, etc.):

*"I: How is the training session programmed?*

*AC1: We all planned the session. Everyone helps in planning the session. We define the exercises according to what we think could be the best option to fit the purpose of the session or the learning content. Then, he* [HC] *decides who leads each drill through the leadership notes. If a specific note is needed, he highlights it. We led that drill but focused on what he highlighted. There are many training sessions where he does not lead any exercises.*

*I: Seriously?! And what do you think about that?*

*AC1: I think it is amazing for everyone. Also, it is a way for him to observe more. [. . .] He gives us all this freedom. We all lead the drills in every training session."*

*#4 AC1*

3.2.4. Reflecting to Understand Their Own Professional Development: The Role of Writing Reflection and Reflective Talks

As demonstrated next, for the HC, reflection skills were a core aspect in the education of their assistant coaches. The development of reflection skills follows a predefined sequence and timing. First, written reflection is encouraged by the reports of training camps, in which all the staff, players, and facilities are assessed. Next, based on the written insights of each member, group reflection unfolds during staff meetings:

*"HC: They have to write the final report of the training camp. James has until Sunday for the report to be on SharePoint. Otherwise, he is in trouble. James must assess each player, the entire staff, the facilities, and a final reflection. How?! Writing! It is not like: 'I think this or that, let's get together and talk'. Okay, debating is important, but write first. Then, we will discuss. [. . .] This series of tasks are the most important educative part."*

*#2 HC*

As demonstrated in the next excerpts, beyond a pedagogical-didactical strategy implemented and promoted by the HC, reflexive talks are already part of the culture of the staff:

*"AC2: Many times, in-season competitions* [talking about the club events], *in our free time, we chat. If the competition starts in the afternoon, we are there in the morning and have meetings all morning. We then have lunch and chat again until the game time."*

*#2 AC2*

*"AC3: For example, a sharing moment is always lunchtime. We usually have an hour for lunch, but it took us two hours because a conversation always comes up depending on what the training was like. What we thought about the practice…It comes naturally."*

*#3 AC3*

As reported, reflection seems to provide meaning to the dilemmas of practice. Also, all the structures beyond the procedures of reflection enabled them to interpret the lived experience, translating it into contextualized knowledge. In this way, reflection about the practice seems to be not only a pedagogical-didactical strategy that supports professional development but also a problem-solving approach:

*"I: And how do you feel that you can develop this reflection and knowledge acquisition process?*

*AC3: Our context of reflection is very privileged. In some way, we try to validate everything that is our behavior through game analysis. Through the repeated analysis of the game. To reflect on the best ways to teach the game, for instance. We reflect on what we think we made more mistakes. And this always ends up being very advantageous […] How can we transform this into knowledge?! By reflecting, it is what validates our beliefs."*

*#4 AC3*

## 4. Discussion

Through an interpretative case study conducted inside a high-level performance context, this study aimed to explore how the head coach of a world champion futsal team worked as a coach developer of their technical staff. Specifically, it was intended to investigate the main pedagogical strategies used, how they were applied, and their impacts on the perceived professional development of the assistant coaches. Overall, the findings demonstrated an intentional structuration of sequence and timing of the tasks and activities assigned by the HC to their ACs. Such tasks are used not only to improve and increase the team's performance but also to develop ACs' professional development. Evidence emphasized the vision of mistakes as learning opportunities, the fostering of commitment of ACs to enhance the team's performance and affording space to plan and lead in practical real-life contexts as the main pedagogical strategies adopted by the head coach (i.e., CD) to promote the ongoing professional development of their assistant coaches. Also, the HC considered reflective skills to be a paramount competency for being a coach, highlighting written reflection and reflective talks as pedagogical tools to develop such competency. The findings demonstrated that from the ACs' perspective, these teaching-learning approaches (i.e., learning by doing and reflecting) and pedagogical strategies have largely impacted their personal and professional growth.

From the perspective of the assistant coaches, the structure of the work built by the HC is portrayed to be extremely important. Indeed, the ACs understood the connection between all pedagogical tasks and their sequence and timing. For instance, the training camp report must be completed and assessed by all the members before the technical team meeting, mirroring the importance of the triad of doing-thinking-interpreting [4]. Moreover, the notion of demand (i.e., the activities are mandatory) was emphasized by all ACs. In this case, the demanding nature of the tasks makes sense since, contextually, everyone shares a committed and accountable aim for raising the team's competitive performance. Thus, it is noted that with a constant and ongoing structure of work, which emphasizes the "doing" portion (i.e., practical experience [4,20]), it was possible to build an environment in which the professional development of coaches occurred and was monitored naturally.

Inside the collaborative working environment, when performing the CD's evaluator role, the head coach considers mistakes to be learning opportunities (i.e., productive failures [42]) instead of weaknesses. This strengthens the supportive agency and meaningful learning from the living experiences [43]. Such positioning provided the assistant

coaches a sense of safety and comfort even during uncomfortable moments (e.g., leading a drill during a training session), which is mandatory to build understanding about their own professional development and coach education process [44]. While performing the educator role, the CD used the assignment of responsibilities (e.g., programming, planning, and leading video sessions for preparing for competition) as a vital pedagogical strategy that entails several goals. From the assistant coaches' perspective, such responsibilities engage them in the process of raising the performance in competition while making them accountable for it. Moreover, it allows each assistant coach to add his or her own and singular vision, enriching and strengthening the team. Still, this naturally promotes experiential and situated learning since the activities and tasks performed occurred inside their daily tasks and activities as coaches [3]. In the head coach's perspective, the attribution of tasks provides data that guide his or her work as a CD, since they enable him or her to make a formative assessment of the professional development of their peers, placing them in the forefront of the adaptation when needed (i.e., "thinking" and reflective experience).

The findings particularly highlighted reflection skills as paramount for the head coach. For instance, ACs must complete a report of training camp that includes a detailed written reflection, followed by reflective talks during staff meetings. In this way, the CD invites his or her staff to first identify and critically interpret the events that occurred and then update and transform the lived experience into knowledge by verbally explaining their thoughts (i.e., "interpreting" and contextual and social experience). Thus, framed by the written reflection, the reflective practice (i.e., the ability to reflect on one's actions to engage in a process of continuous learning [45]) is addressed through reflective conversations. Thereby, the CD used first the written reflection to systematize the thoughts and interpretations and then the reflective talks to verbalize and share different insights, which is crucial to developing new knowledge [19]. The staff acknowledged this intentional procedure and mentioned its importance in stimulating their professional growth, enriching the teaching-learning process as well. This aligns with previous studies that demonstrated how structured narratives and reflexive conversations are important tools to develop coaches' understanding of their practice (e.g., [46,47]). In this way, the CD also assumes a role as a facilitator of learning [14].

The term "facilitator of learning" [48] derives from the scaffolding framework [49], which refers to the temporary and structured support given by the CD to the ACs when performing a task (e.g., planning a drill) that may otherwise not be achieved alone. Thus, by attributing challenging activities, tasks, and responsibilities in which the support provided is tailored according to assistant coaches' needs, the head coach scaffolds his or her peers' personal and professional development. The notion of CDs as learning facilitators was previously highlighted in the study by Culver et al. [50], and our findings extend the operationalization of how to address coaches' learning needs, endorse reflection, and engage all the actors involved.

## 5. Final Thoughts and Practical Applications

This interpretative case study portrayed how a head coach could work as a CD inside his or her technical teams, revealing essential pedagogical strategies such as structuring the sequence and timing of the tasks and activities suggested, using prompts for reflection, assigning ACs with diverse tasks, promoting their accountability, and creating a trusting space where failing is considered a learning opportunity (Figure 1). Moreover, the investigation depicted how such pedagogical strategies are vital to the personal and professional development of the assistant coaches. Although this is an examination of a unique reality, this investigation offers relevant practical insights. First, given its broad application, the strategies reported could be used in similar educational contexts of intervention. Second, the findings implicitly demonstrate how the assistant coaches' role has changed throughout the years, from unknown workers to recognized and valued members of the technical team's development. Since the leading role of the assistant coach has been raised in practical contexts, future investigations may be dedicated to investigating its

role, professional competencies, and personal features. Third, as coach education acquires a personal meaning when it is guided by an expert (i.e., a significant peer), the sportive federations could work on the continuous professional development of their coaches inside their own sports contexts, meaning developing experienced head coaches as CDs so that the training process can occur locally (i.e., in the clubs) inside each singular practice niche.

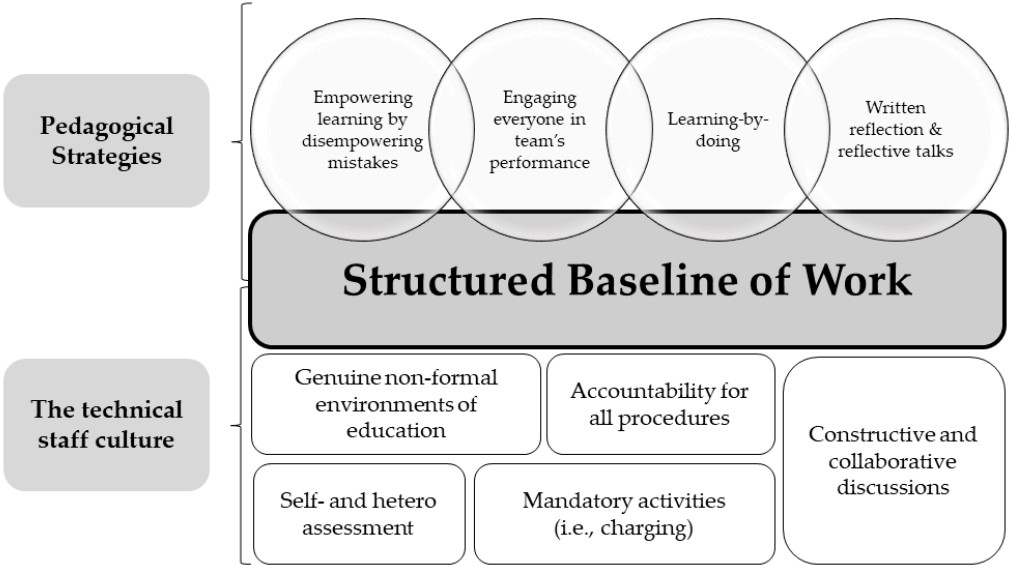

**Figure 1.** An example of good practices to stimulate the ongoing education and professional development of coaches.

**Author Contributions:** Conceptualization, A.G.R. and I.M.; methodology, A.G.R. and C.V.; formal analysis, A.G.R., C.V. and I.M.; writing—original draft preparation, A.G.R.; writing—review and editing, C.V. and I.M.; All authors have read and agreed to the published version of the manuscript.

**Funding:** This research received no external funding.

**Institutional Review Board Statement:** The study was conducted in accordance with the Declaration of Helsinki and approved by the Institutional Research Ethics Committee of the Faculty of Sports, University of Porto (CEFADE 50 2022).

**Informed Consent Statement:** Informed consent was obtained from all subjects involved in the study.

**Data Availability Statement:** No new data were created or analyzed in this study. Data sharing is not applicable to this article.

**Conflicts of Interest:** The authors declare no conflict of interest.

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
