# Peer review of "The Head Coach as a Coach Developer: A Coach Education Case Study inside a World Champion Futsal Team"

_education, doi:10.3390/educsci13121229_

Round 1

Reviewer 1 Report

Comments and Suggestions for Authors

The title reflects an interesting topic, current in the current stage, which is of vital importance

I consider the introduction to be correctly elaborated, correctly structured, the statements are appropriately selected and reinforced by bibliographic sources;

Materials and methods: clarity and optimal approach, but in applying the questionnaire I would mention some suggestions that will also improve the content of the Results chapter.

In this sense, I recommend the authors to try a statistical processing of the answers, even if we are talking about a case study, or to apply the respective questionnaire in 2 different moments to be able to make various comparisons.

In addition, a general (orientative) profile or guide of good practices regarding the development/education of the coach (general principles) could be elaborated.

Discussions: to be completed depending on what you add to the Results part.

The bibliography: includes an interesting mix of sources, in which bibliographical sources newer than 10 years predominate.

Reviewer 2 Report

Comments and Suggestions for Authors

The review for this scientific article highlights an in-depth analysis of coach education, focusing on how the head coach (HC) of a world champion futsal team acts as a developer of assistant coaches (AC).

The research method, an interpretive case study, proves to be suitable to investigate the complexity and uniqueness of technical staff in the educational context. Here is the only objection I make, namely the small number of subjects involved in the research.

The conclusions are relevant and underline the vision of mistakes as learning opportunities, stimulating commitment and improving team performance as well as giving space for planning and leadership in practical contexts.

The perspective of CAs emphasizes the significant impact of these strategies on personal and professional growth. The results suggest that the study could provide useful guidance for the redesign of future coach education programs.

Congratulations. All the best!.

Reviewer 3 Report

Comments and Suggestions for Authors

The data analysis of the interviews are not sufficient enough, I would suggest more individual coders (at least 3 or four of them) to recode and do the content analysis. These coders must not just individually work on the interviews but independently, preferably out of the authors. Even using content analitical programmes, or sofwares (such as ATLAS) should be used for the valid content analysis. Otherwise, I can not accept this work on this state, so further analyses are needed.

Round 2

Reviewer 1 Report

Comments and Suggestions for Authors

Is ok now